# Lead-Free Metal Halide Perovskites for Hydrogen Evolution from Aqueous Solutions

**DOI:** 10.3390/nano11020433

**Published:** 2021-02-09

**Authors:** Vincenza Armenise, Silvia Colella, Francesco Fracassi, Andrea Listorti

**Affiliations:** 1Department of Chemistry, University of Bari “Aldo Moro”, via Orabona 4, 70126 Bari, Italy; vincenza.armenise@uniba.it (V.A.); francesco.fracassi@uniba.it (F.F.); 2CNR NANOTEC Institute of Nanotechnology, Via Amendola, 122/D, 70126 Bari, Italy; silvia.colella@nanotec.cnr.it

**Keywords:** metal halide perovskites, lead-free, photocatalysis, solar-driven hydrogen evolution, aqueous solutions, water-stable, materials properties

## Abstract

Metal halide perovskites (MHPs) exploitation represents the next big frontier in photovoltaic technologies. However, the extraordinary optoelectronic properties of these materials also call for alternative utilizations, such as in solar-driven photocatalysis, to better address the big challenges ahead for eco-sustainable human activities. In this contest the recent reports on MHPs structures, especially those stable in aqueous solutions, suggest the exciting possibility for efficient solar-driven perovskite-based hydrogen (H_2_) production. In this minireview such works are critically analyzed and classified according to their mechanism and working conditions. We focus on lead-free materials, because of the environmental issue represented by lead containing material, especially if exploited in aqueous medium, thus it is important to avoid its presence from the technology take-off. Particular emphasis is dedicated to the materials composition/structure impacting on this catalytic process. The rationalization of the distinctive traits characterizing MHPs-based H_2_ production could assist the future expansion of the field, supporting the path towards a new class of light-driven catalysts working in aqueous environments.

## 1. Introduction

Nature, through plant photosynthetic processes, demonstrates the viability of directly converting sunlight into fuels, which involves the storing of the energy from the incident solar irradiation in the form of chemical bonds [1]. We benefit from such long-lasting processes to power our ever-growing energy needs, even if the consequences of an out of equilibrium utilization of such resources is threatening the very fabric of our society. Today, more than the 80% of total energy supply comes from thermoelectric conversion of fossil fuels leading to harmful environment alterations [2]. The desirable modification of such scenario involves a large use of renewable energies, both for the direct production of electricity (solar, wind, hydroelectric, etc.) and for the generation of green fuels.

Hydrogen (H_2_) have been envisaged as one of the most promising energy vectors, due to its high energy content, the highest among any common fuel by weight [3] but also because it can be consumed to produce energy releasing only water as reaction outputs [4]. At present, H_2_ production relies on fossil fuels and it is mostly based on methane steam reforming [3,5]. Alternatively, it can be produced via electrolysis and the electricity used to drive such process can come from photovoltaic (PV) devices or wind turbines, reducing the reliance on fossil fuels. The straightforward approach on this line is the water splitting in which semiconductor materials are submerged in aqueous solution to directly split water into oxygen (O_2_) and H_2_ using sunlight [6,7]. The capability of the semiconductor to efficiently run the reaction, to sustain numerous cycling and, ideally, to be derived from eco-friendly and economic manufacturing procedures is central in this promising approach.

In the last decade, a newcomer in the field of semiconductors for optoelectronics has raised great interest as potential next generation reference material. In fact, metal halide perovskites (MHPs) have revolutionized the field of emerging PV reaching in few years the status of silicon best competitor, impacting as well on light emitting and detector technologies [8]. This disruptive effect is a direct consequence of the superior properties of such hybrid semiconductors, namely high optical absorption in the visible region, tunable band-gap, high defects tolerance leading to long carrier lifetimes, withstanding straightforward deposition processes at mild conditions [8]. The easy tunability of their electronic band structure potentially places the band edges of many MHPs in excellent positions to perform also photocatalytic reactions, such as the above cited H_2_ generation, the carbon dioxide reduction and the organic dye degradation under visible-light irradiation [9]. However, these photocatalytic processes, which are among the most interesting ones, often require the use of water or alternative polar solvents, representing a conundrum foreseeing MHPs utilization. It is well known, in fact, that MHPs are highly unstable upon moisture and water exposure, therefore those media are hardly compatible with most of the currently known MHPs [9]. Despite these intrinsic constraints, different strategies have been used to overcome the issue, in particular three main paths have successfully been exploited [9]. The first one (Figure 1a) foresees the encapsulation of the active perovskite layer with barriers capable of avoiding water exposure but allowing carrier flow. Field’s metals and carbon derivates were successfully used for the task [10]. This approach allowed the inclusion, for the first time, of MHPs in complete photoelectrochemical (PEC) cells. Herein, only lead-based (Pb-based) perovskites have been used so far, although there are not intrinsic obstacles for the use of lead-free MHPs in such kind of devices. The second strategy, used for the first time in 2016, foresees the exploitation of complex dynamic equilibria between perovskite precursors and perovskite powders dispersed in solution (Figure 1b) [11]. In order to sustain the continuous formation of suitable perovskite crystalline phases, a high concentration of hydrohalic acids (e.g., HI or HBr) has to be present in solution [9,12,13,14]. Indeed, their dissociation, leading to high halogen concentration, shifts the solubilization equilibrium towards the formation of MHPs. In addition, the hydrohalic acid can directly give the reagents for H_2_ formation in a particular mechanism in which the halide ions function as hole scavenger. The last strategy implies the use of water-stable MHPs [9,14] (Figure 1c). The very recent conception of such materials allowed the first seminal utilization of MHPs directly in water for the solar-driven H_2_ evolution [15,16,17].
(1)$$\;\mathrm{Perovskite}\;\to e^{-}+h^{+}$$
(2)$$2e^{-}+2H^{+}\to H_{2}$$
(3)H2O+h+→12O2+H+
(4)$${\mathrm{RCH}}_{2}\mathrm{OH}+2h^{+}\to \mathrm{RCHO}+2H^{+}$$
(5)$$3I^{-}+2h^{+}\to I_{3}^{-}$$

The use of MHPs in aqueous solution gives serious concerns about the presence of Pb in the most performing ones. In fact, atoms of this element in the perovskite structure, particularly for their outermost s electrons with a 6s^2^ electronic configuration, represent one of the key ingredients for the obtainment of the material superior optoelectronic properties, especially the one related to charge dynamics [18,19]. At the same time, the use of Pb salts in the material manufacture can have a dramatic impact on the environment and on human health [20,21]. Indeed, Pb is a toxic metal, which accumulates in the vital organs of men and animals and it enters in the body through air, water and food. It can cause cumulative poisoning effects, such as serious hematological damage, anemia, kidney malfunctioning and brain damage. Moreover, chronic exposure to Pb causes severe lesions in kidney, liver, lung and spleen. This aspect is already an obstacle foreseeing large scale manufacturing of perovskite-based photovoltaic devices but is even more pressing for photocatalytic application [22]. In fact, in this case an intimate contact of the active material and the reaction media is present, thus in conditions where leakages into the ambient could easily happen. Related to this central issue, a recent work by Li et al. showed how the Pb from perovskite, leaking into the ground, can enter plants and consequently the food cycle ten times more effectively than other Pb-based contaminants already present as the result of human activities [23].

For these reasons and with the aim of deepening the study on the MHPs application for the H_2_ evolution in aqueous solutions, we believe it is really important to focus on lead-free (Pb-free) materials at the initial stage of the research in this field. Thus, this minireview is centered on the application of Pb-free MHPs for H_2_ evolution from aqueous media, whose performances, measurement conditions and references are collected in Table 1 preceding the concluding remarks. Particular attention is placed here on the relationship between material structure/composition and reaction efficiency, by which we depict some potential strategies for the future development of the field. The work is organized in different sections: a first one in which the mechanisms and the possible reactions for the H_2_ evolution involving Pb-free perovskites in aqueous solutions are introduced and discussed, as well as the figures-of-merit for comparing different systems; the main section that deals on the different Pb-free perovskite materials successfully used for the task, followed by a conclusive and perspective part.

## 2. General Mechanisms and Figures of Merit

In the process of H_2_ evolution, the photocatalytic system requires reactants, photocatalyst, photoreactor and light source. In details, the following steps are involved: (1) light harvesting, (2) charge formation, (3) charges separation and transfer and (4) catalytic reactions on the photocatalyst surface or redox reaction (Figure 1d).

Firstly, incident light with energy greater than or equal to the band gap (E_g_) of the photocatalyst strikes over the perovskite surface. For most of the MHPs-based photocatalytic systems, visible-light is efficiently absorbed. Secondly, when light strikes over the perovskite, electronic transitions are initiated, resulting in generation of electron/hole couples (Equation (1) in Figure 1d). Thirdly, electrons and holes are separated, due to the excitation of electrons from the valence band (VB) to conduction band (CB), leaving holes in the VB. Lastly, redox reaction takes place on the surface of perovskite photocatalyst when the reduction potential of perovskite is above the CB and oxidation potential is below the VB of the photocatalyst [24]. Within the redox process, electrons are involved in converting protons (i.e., H^+^) into H_2_ over the surface of photocatalyst (Equation (2) in Figure 1d) [25], while holes are involved in the oxidation of water (Equation (3) in Figure 1d) or of a sacrificial agent (Equation (4) in Figure 1d). Indeed, this depends on the media in which the reaction takes place: it can be pure water, water mixed with a sacrificial agent or an acidic solution of HI or HBr (Figure 1a–c).

Photocatalytic water splitting is a challenging task because the entire process starting from charge carrier generation to surface catalytic reactions for producing H_2_ requires appropriate light energy and intensity. In fact, the reaction requires energy (∆H = 286 kJmol^−1^) and multiple electron transfer steps [26]. However, most challenging task is the fast recombination of photogenerated charge carriers due to short lifetime. Electrons generated in the CB can recombine with the holes inside the volume of photocatalyst or at the surface (i.e., volume and surface recombination), to generate unproductive energy as a heat [27]. Therefore, for successful water splitting, the redox reaction and charge separation in the photocatalyst must proceed within the lifetimes of the photo-excited charge carriers [28]. Herein a prominent role is played by sacrificial agents or electron donors/hole scavengers such as alcohols, triethanolamine (TEOA) and sodium sulfide/sodium sulfite which are the most typically used to capture the photo-generated holes with great oxidizing properties, allowing the H^+^ reduction to proceed more smoothly. An example of alcohol oxidation is illustrated in Equation (4) of Figure 1d. Most of the photocatalytic systems reported in this review are characterized in the presence of TEOA as sacrificial agent (Table 1).

It is worth noticing that, at this early stage of research on Pb-free catalytic systems, most of the developed perovskite photocatalysts are not intrinsically stable in water but, as previously mentioned, they are stabilized as particulate in a hydrohalic solution, usually of HI or HBr. Generally speaking, by controlling the I^−^ or Br^−^ and H^+^ concentrations, powdered perovskites undergo a continuous dissolution and precipitation under a dynamic equilibrium, which allowed a stable photocatalytic HI or HBr splitting and evolution of H_2_ as of I_3_^−^ (Equation (5) in Figure 1d) or Br_3_^−^. As can be simply observed in Table 1, H_3_PO_2_ is often added to the hydrohalic solution with the purpose to reduce I_3_^−^ or Br_3_^−^ generated during the HI or HBr splitting, that would otherwise interfere with light absorption [11,29,30,31,32,33]. Only a few perovskite systems have been reported up to now to possess an intrinsic stability in water, namely DMASnBr_3_ [17,34], DMASnI_3_ [15] and PEA_2_SnBr_4_ [16], which are extremely promising for future developments.

The performances of a system for H_2_ evolution can be quantified in different ways, depending also on the analyzed system.

The most complete figure of merit is the Solar-to-Hydrogen conversion efficiency (η_STH_), expressed for a full electrochemical system comprising two electrodes, such as the one sketched in Figure 1a. It is generally defined as the amount of chemical energy (stored in H_2_) produced against the incident solar energy [35], by which all H_2_ evolving devices can be reliably compared [35,36,37,38,39,40,41]. By using standard solar irradiation, for instance generated with the Air Mass 1.5 global (AM 1.5 G) filter or one sun (100 mW cm^−2^), with no applied biased [35] the Equation (6) can be applied to determine it.
(6)ηSTH=[Chemical energy producedSolar energy input] = [Rate of H2 production × ΔGH2O→ H2 + 12 O2 Total incident solar power × Electrode Area] = [(mol H2 per s) × (237,000 J mol−1)PTotal(W cm−2) × Area (cm2)] AM1.5G

The rate of H_2_ production can be directly measured by gas chromatography or mass spectrometry as mmol cm^−2^ h^−1^ for the PEC systems. Alternatively, it can be determined from the generated photocurrent density (J_sc_) as follows:(7)$$\eta_{\mathrm{STH}}=\left[\frac{J_{\mathrm{sc}\;}\left({\mathrm{mA}\;\mathrm{cm}}^{-2}\right)\;\times \left(1.23\;V\right)\;\times \eta_{F}}{P_{\mathrm{Total}}\left({\mathrm{mW}\;\mathrm{cm}}^{-2}\right)}\right]{\;}_{\mathrm{AM}1.5G}$$
where η_F_ is the Faradaic efficiency of H_2_ production, that is, the efficiency with which charges contribute to H_2_ evolution reaction.

For the particulate systems, where a powder is dispersed in aqueous media (e.g., Figure 1b,c), the quantification is much more straightforward and much less standardized. Very often the comparison is made among the rate of H_2_ production. Particularly, the photocatalytic activity for most of the perovskite-based systems, developed at this preliminary stage of development and discussed in this review (Table 1), is reported in mass unit per time, that is, µmol g^−1^ h^−1^.

If the measurement is normalized for surface and light irradiance, an apparent quantum efficiency (AQE) can also be calculated according to Equation (8), where the photon flux of the incident light is determined using a spectroradiometer [32,42].
(8)$$\mathrm{AQE}\;\left(\%\right)=\left[\frac{2\times {\;\mathrm{number}\;\mathrm{of}\;\mathrm{evolved}\;H}_{2}\mathrm{molecules}}{{\mathrm{number}\;\mathrm{of}\;\mathrm{incident}\;\mathrm{photons}}_{}}\right]\times 100$$

It is worth noting that light irradiation should ideally be standardized and widely accepted within the community, however the measurements conditions employed are still very scattered. Other important information that have direct impact on the photocatalytic performance and which are often missing in the reviewed works, are the one related to the morphology/dimension of the particulate. Those features should be ideally highlighted in future works due to their impact on the MHPs properties. On one hand, crystal dimension influence material optics, on the other hand morphology determines the surface exposed to the reaction thus the photocatalyst activity.

## 3. Pb-Free Perovskites for H_2_ Evolution

MHPs have the general chemical formula ABX_3_, where A is a monovalent organic or inorganic cation (e.g., MA^+^ = methylammonium, FA^+^ = formamidinium, DMA^+^ = dimethylammonium, Cs^+^), B is a metal cation (e.g., Pb^2+^, Sn^2+^, Ge^2+^) and X is a halide ion (e.g., Br^−^, I^−^, Cl^−^) [9,12,43]. In a typical perovskite structure a cubic three-dimensional (3D) network in which the B cation coordinates with six X sites to form [BX_6_]^4−^ octahedra, is neutralized by the A cations filling the 12-fold generated cavities counterbalancing the charge of the extend inorganic framework [12,44,45] (Figure 2a). Generally, the cation that occupies the A-site is defined the responsible for the orientation cage within the 3D structure, affecting its symmetry [46]. It has no direct contribution toward the electronic properties. However, its size can cause the tilting of [BX_6_]^4−^ octahedra as well as the strain of the B-X bonds which control the above cited properties [13,46,47]. Indeed, both the B- and X-site play a relevant role in constructing the MHP band structure, fundamental for the electronic and optical properties of this material [13,46,48]. The B-site is typically occupied by metals that belong to the IVA group of the periodic table of elements, in a divalent oxidation state. Upon proceeding up in the IVA group (i.e., from Pb to Ge) a variation of the optoelectronic properties and structure can be observed, a reduction in stability of the divalent oxidation state is verified as an effect of reduced inert electron pair effects. This also corresponds to an enhancement of its electronegativity, which implies a reduced difference with respect to the halogen, turning in a decrease of the material band gap [46,47]. With respect to the X-site upon proceeding down in the VIIA group (i.e., from Cl to I), the ionic size increases and the MHP absorption spectra shifts to longer wavelengths as well as to lower energy (i.e., red-shift). This effect can be attributed to the reduced ionic and increased covalent character of the bond [46,47].

Considering all ions as rigid spheres, the formation of a stable ABX_3_ perovskite is related to the size of its ions, as described by the Goldschmidt’s tolerance factor in the Equation (9):(9)$$t\;=\;\frac{r_{A\;}+{\;r}_{X}}{\surd 2\;\left(r_{B}+{\;r}_{X}\right)}$$
where r_A_, r_B_ and r_X_ are the radii of A, B and X ions, respectively [49,50]. Particularly, it was demonstrated that the t factor should be in the range 0.8–1.0 for 3D cubic frameworks, with a t value of 1.0 indicating an ideal one [12,13,44,45,47,49]. In addition, the semiempirical geometric parameter reported in Equation (10) can be utilized to predict the stability of the octahedral structure.
(10)$$µ\;=\;\frac{r_{B\;}}{r_{X}}$$

Its value should be in the range 0.4–0.9 for stable [BX_6_]^4−^ [12,13]. Although it is important to underline that the perovskites stability can be affected by external parameters such as illumination, thermal stress, exposure to ambient air, moisture and the previously mentioned immersion in water or other solvents as solutions, which can cause the collapse or the transformation of the crystalline network [12,51]. Specifically, the detrimental action of water on perovskites structures is well known and can be summarized by the degradation pathway of the referential 3D MAPbI_3_ one, reported in Equation (11), that leads to the intermediated (MA)_4_PbI_6_ × 2H_2_O hydrated phase as PbI_2_.
4 MAPbI_3_ + 4H_2_O → 4[MAPbI_3_xH_2_O] → (MA)_4_PbI_6_x2H_2_O + 3PbI_2_ + 2H_2_O (11)

In particular, it has been proved that the degradation pathway is triggered by the water molecule entrance in the cavity hosting the organic cation (i.e., MA^+^) following a nucleophilic attack of the halogen, which causes the reversible formation of the monohydrate phase from here the reaction path foresees the release of one of the material precursors and the collapse of the structure. The nature of the organic cation but, also, the strength of the bond between the central bivalent cation and the halide, have therefore a huge impact on these degradation path. Logically hydrophobic cations could protect the structure from the water entrance and strong metal halide bonds could offer superior resistance to the attack [52].

By introducing a large cation in the A-site of the 3D ABX_3_ structures, those become instable and a transformation of the materials towards low-dimensional (i.e., two-, one- or zero-dimensional) species can be triggered [12,44,45,50]. The low-dimensional MHPs derivatives can be obtained by cuts or slices along different crystallographic directions in ABX_3_, in those systems [BX_6_]^4−^ octahedra are no longer corner-shared [12,44]. Among them, two-dimensional (2D) perovskites consist of multiple or single inorganic sheets sandwiched between bulky organic cations (i.e., spacers) held together by Coulombic forces (Figure 2b) [44,51,53,54]. The layered framework enables the organic spacers cations to be accommodated between inorganic sheets and it relaxes the limits on the cation sizes imposed by the Goldschmidt’s tolerance factor. Therefore, a wide choice of these cations it is possible, as they can be designed to attain desired perovskite features [53,54]. Additionally, a pure 2D structure is formed when single layers of the inorganic framework are isolated by the bulky cations, while multi-layered structures are created by increasing the number of these layers and they converge to 3D structures for an infinite number of them [51,53]. One-dimensional (1D) perovskites consist of linear [BX_6_]^4−^ octahedra chains, whereas zero-dimensional (0D) ones are compounds in which octahedra are singularly isolated (Figure 2c) or generate isolated dimers with a face-shared and can easily shift in relative position [44,55,56]. The dimensional reduction has a significant impact on the MHPs properties. For instance, the band gap typically increases as the dimensionality of their structure is lowered and the stability of low-dimensional materials is higher than that of the 3D ones [13,44,51]. Consequently exciton binding energies and carrier transport capabilities are also modified by the dimensionality reduction, with the former increasing as dimensionality is reduced due to dielectric confinement, while conduction become directionally dependent due to the non-continuous inorganic framework distribution [53].

Low-dimensional halide perovskite derivatives can also be obtained by replacing in ABX_3_ the divalent metal cation of the B-site with trivalent cations (e.g., Sb^3+^, Bi^3+^) [57,58,59]. In fact, due to their +3 oxidation state these cations are not able to form a corner-sharing octahedra 3D continuous structure, therefore a change from ABX_3_ to A_3_BX_6_ or A_3_B_2_X_9_ is observed and these latter have 0D or 2D structures with strongly bound excitons and low carriers mobilities which are the main causes for the poor efficiencies of A_3_B_2_X_9_ perovskite-based solar cells [57].

Beyond the earlier mentioned MHPs, double ones have been also reported in literature [57,60,61]. They can have the general formula A_2_B’B”X_6_ in which A is a monovalent cation, B’ and B” are a pair of heterovalent (i.e., monovalent and trivalent) metal cations to give an average +2 valence state (e.g., Ag^+^ and Bi^3+^), while X is a halide [60,61,62]. These kind of perovskite derivatives provide an avenue for easy substitution and incorporation of metal cation in B’- and B’’-site, the possibility to use various organic or inorganic species in the A-site and a variation of the halide composition in the X-site [60]. Ideally, their structure is very similar to that of ABX_3_ perovskites, with [B’X_6_]^4−^ and [B”X_6_]^4−^ octahedra formed by B’ and B” cations coordinated with six X anions, respectively and A cations occupying the octahedral cavities (Figure 2d). An alternating arrangement of [B’X_6_]^4−^ as [B”X_6_]^4−^ octahedra can be observed, known as rock salt ordering [60,61]. Another general formula of the double perovskites can be ABX_6_, in which the B-site is occupied by a tetravalent cation as a vacancy while A and X have the same features of the other cited materials [57].

Therefore, the immense number of composition and structural variation implies the generation of a plethora of MHPs and their derivatives, with wide different properties such as E_g_, exciton binding energy, carrier’s mobility and effective mass of the carriers. Envisioning MHPs utilization as photocatalysts, based on the reaction thermodynamics, a suitable match is required between the electronic band structure of these semiconductor and targeting reactions redox potential. The potentials of typical photocatalytic half-reaction involved in water splitting are shown in Figure 3, alongside the relative position of CB and VB of some representative MHPs. In the Figure some of the traditional photocatalysts, namely TiO_2_ and Fe_2_O_3_ are also reported [7,9]. Clearly, many other examples of efficient alternative photocatalytic materials could be found in literature, (e.g., metal dichalcogenides and oxides [63] as well as hybrid structures [64,65,66]) however the discussion of those alternatives lies outside the focus of this minireview. Figure 3 highlights the excellent tunability of Pb-free MHPs properties, in particular the water reduction ability related to the relative position of their CB; typically, negative enough for H_2_ generation. Moreover, other MHPs (e.g., chloride- or bismuth-based) materials have the possibility to even oxidize water, thus suggesting a route for full water splitting by MHPs. Although strong efforts are placed in the research front, Pb-based perovskites are hardly replaceable for photovoltaic exploitation due to a combination of unique, highly advantageous properties [57,59]. The MHPs features required for H_2_ photocatalytic evolution from aqueous solution can be somehow different, allowing the exploration of a wide range of materials. Among the key needs, one should include for sure stability under operative conditions, preventing the structure collapsing after the contact with the solutions, allowing for highly process cyclability and avoiding environment contamination. For these reasons, as previously mentioned, the use of Pb-free MHPs needs to be wholeheartedly pursued. Thus the few examples of lead free MHPs used so far for the H_2_ generation in water and/or in its solution will be discussed and classified according to their chemical composition and structure as follows: (i) 3D tin-based (Sn-based) MHPs, (ii) 2D Sn-based MHPs, (iii) 0D bismuth-based (Bi-based) as antimony-based (Sb-based) MHPs and (iv) double MHPs.

### 3.1. 3D Sn-Based MHPs

3D Sn-based MHPs have the general chemical formula ASnX_3_ in which Sn is an environmentally friendly element of the IVA group with an ionic radius comparable to that of Pb (Sn^2+^ 1.35 Å and Pb^2+^ 1.49 Å) [47,58]. These materials possess properties similar than Pb-based ones, with narrower band gaps and potentially higher carriers mobilities. However, Sn^2+^ cation is only relatively stable, as the most common oxidized state is the tetravalent one, reason why they are usually instable under exposure to ambient air, moisture and/or water, limiting thus their practical applications [58,59].

To address this stability problem and to make these 3D perovskites suitable for the H_2_ production in water solutions, Pisanu et al. [34] synthesized bulk powder samples of MA_1−x_DMA_x_SnBr_3_ perovskite in which x varies in the range 0–1, that reflected in a clear and progressive color change from intense purple (for x = 0) to off-white (for x = 1). MASnBr_3_ sample has a cubic crystal structure, by introducing DMA into the MASnBr_3_ lattice a cubic cell is retained up to x = 0.60, above this value samples are a mixture of cubic and orthorhombic phases as DMASnBr_3_ has an orthorhombic distorted structure (Figure 4a). The replacement of DMA for MA leads to a blue-shift of the absorbance edge, from about 2.0 eV for x = 0 to 2.9 eV for x = 1. Aging tests combined with X-ray photoelectron spectroscopy (XPS) analyses demonstrated that DMASnBr_3_ is highly tolerant to air-exposure in terms of both crystal structure as optical properties representing an intriguing example of highly stable ASnX_3_ perovskite. Stability could be related to the presence of large organic hydrophobic cations in the 3D structure, to the hydrogen bonding network and/or related to the electronic structure [34,53,59]. The effects of the immersion into deionized (DI) water on DMASnBr_3_ samples was were subsequently tested [34]. The formation of an opalescent solution without the dissolution of the perovskite was observed and preliminary tests for H_2_ production were performed. 2 g L^−1^ of DMASnBr_3_ were suspended in DI water, the suspension was deoxygenated by Ar, then it was irradiated under simulated solar light, using a Solar Box (500 W m^−2^) with an UV outdoor filter, for 5 h and the evolved gas was quantified by gas chromatography coupled with thermal conductivity detection (GC-TCD). A H_2_ photoproduction of about 6 µmol g^−1^ h^−1^ was observed [34]. An additional experiment was performed in the presence of 10% *v*/*v* of TEOA as typical sacrificial agent and 1 wt% of Pt, which acts as co-catalyst leading to a more efficient separation of generated charges [11,42,67] and a H_2_ production of 11 µmol g^−1^ h^−1^ was measured [34]. These performances were quite low respect to the most active photocatalysts for this application but represented a very encouraging seminal achievement in this innovative field of the research [14,46,68,69].

Aimed to improve the H_2_ evolution reaction performance of water-stable DMASnBr_3_ perovskite, Romani et al. [17] developed a novel approach based the synthesis of composites between the MHP and graphitic carbon nitride (g-C_3_N_4_). The latter is a well-known photocatalyst, acting under visible-light, characterized by a conduction and valence band positions suitable for oxidation and reduction processes as well as a relatively narrow band gap (2.7 eV), this allows also the favorable band alignment with the DMASnBr_3_ (Figure 4b) [16,17,70]. The composites [17] were synthesized by the ball-milling method [71] at MHP loading from 1 to 33 wt% and exhibited an overall amorphous-like structure. H_2_ evolution experiments demonstrated an efficient photocatalytic activity for the composite based on a 3D lead-free MHP in water. Specifically, these experiments were firstly carried out by suspending samples in distilled water containing 10% *v*/*v* of TEOA, with deoxygenation by Ar and solar light irradiation using a Solar Box (500 W m^−2^), equipped with an UV outdoor filter, for 6 h. Reported results showed that the composites performances are significantly better than g-C_3_N_4_ alone and the highest H_2_ evolution rate (19 µmol g^−1^ h^−1^), measured by means of gas chromatography coupled with GC-TCD, was obtained with a MHP amount of 33 wt%, corresponding to a 10-fold improvement with respect to the pure polymeric material (Figure 4c) [17]. By including 3 wt% of Pt, the 33 wt% MHP composite reached an impressive H_2_ production of 1730 µmol g^−1^ h^−1^. To assess the positive effect of the TEOA as sacrificial agent, the 33 wt% MHP sample with Pt was also tested in pure water and an H_2_ production of 14 µmol g^−1^ h^−1^ was obtained. H_2_ evolution tests carried out in distilled water containing glucose (0.1 M), as biomass-derived sacrificial agent, demonstrated that the same MHP composite in presence of 3 wt% Pt shows a H_2_ evolution rate of 300 µmol g^−1^ h^−1^, 100-fold improved with respect to the pure g-C_3_N_4_ [17].

A similar approach foreseeing the use of DMA cation was also proposed by Ju et al. [15]. They prepared DMASnI_3_ single crystals by means of a temperature lowering method and observed an interesting reversible band gap narrowing with excellent water phase stability. Particularly, the as synthesized perovskite crystals exhibited a rod-like yellow morphology with a length of about 0.3 mm, an orthorhombic crystal system and an E_g_ value of 2.5 eV. Once the exposure to air they turned to black with a band gap of 1.3 eV keeping their morphology as well as crystalline phase unchanged, after wetting with distilled water they self-heal from black to yellow. This peculiar behavior was explained with the formation of iodide vacancies that can lead to the band edge reconstruction as the band gap decrease [15,72]. Moreover, since the trap-state density increases, the single crystal transform into poly-crystals and the treatment with water allows the crystals to recover to their original state. Remarkably, DMASnI_3_ samples remained stable [15] when they were immersed in distilled water for 16 h and presented long carrier lifetimes (τ_1_ = 22 ns, τ_2_ = 495 ns), showing a potential as photocatalyst for H_2_ evolution in water rather than in hydrohalic acid solutions [9,14]. Indeed, photocatalytic H_2_ evolution measurements performed on perovskite powder immersed in distilled water, irradiated with a 300 W Xe lamp for 5 h at a reaction temperature of 10 °C, showed a H_2_ evolution rate, quantified using a gas chromatograph, of 3 µmol g^−1^ h^−1^ with good recycling properties (4 cycles tested) [15]. In this case due to the formation of O_2_ after the evolution of H_2_, the peaks related to the formed SnI_4_ were detected by X-ray diffraction (XRD).

### 3.2. 2D Sn-Based MHPs

The intrinsic instability of 3D MHPs to air, moisture and/or water exposure has been addressed in solar converting devices by introducing large hydrophobic cation, which break the inorganic network reducing their dimensionality to 2D structures. Noticeably these 2D perovskites were the first to be exploited in optoelectronics by Mitzi in the 80s [73]. They have the general formula R_2_A_n-1_B_n_X_3n+1_ where n indicates the number of inorganic layers held together, R is the bulky organic cation and A, B and X are the typical monovalent cation, the metal cation and the halide, respectively [16,53,54]. The bulky R species can occupy the surface sites and protect the inorganic layers from water thanks to their hydrophobic nature [53,74], therefore they can be considered suitable for applications in water.

In this respect, Romani et al. [16] reported a novel Pb-free 2D perovskite, namely PEA_2_SnBr_4_, having the requisite of water stability with suitable optical properties for the photocatalytic applications and no environmental concern. The PEA_2_SnBr_4_ bulk material was synthesized by a wet-chemistry method and it presented an orthorhombic crystal structure. The measured E_g_ was around 2.7 eV and the positions of the valence band minimum (VBM) as of the conduction band minimum (CBM) appeared suitable for the visible light photoinduced H_2_ generation in water. The water stability was evaluated by dispersing the powder in DI water under stirring, recovering them by filtration and measuring the concentration of tin in the water after the perovskite removal (i.e., leaching test). The XRD pattern of samples after 4 h of immersion in DI water, the XPS results (Figure 5a), the absorbance spectra as well as the leaching test confirmed the extraordinary perovskite stability and the absence of any Sn oxidation. H_2_ photogeneration experiments, carried out by immersing the PEA_2_SnBr_4_ in DI water containing 10% *v*/*v* of the sacrificial agent TEOA and 3 wt% of the co-catalyst Pt, with deoxygenation by Ar, under simulated solar light using a Solar Box (500 W m^−2^) provided with a UV filter for 6 h, showed that the perovskite evolves 4 µmol g^−1^ h^−1^ of H_2_, measured by means of gas chromatography coupled with GC-TCD. In order to obtain a more efficient photocatalyst, the authors investigated the creation of composites with the previously cited g-C_3_N_4_ [17]_._ In fact, it owns a band gap that well matches with that of PEA_2_SnBr_4_ forming a heterojunction that reduces the recombination rate of the photogenerated electron-hole pairs (Figure 5b). Particularly, composites were synthesized at MHP loading of 5 or 15 wt% by means of wet-chemistry and exhibited an overall amorphous-like structure with the possible formation of small perovskite particles on the carbon nitride. The H_2_ evolution rates, determined in the same experimental conditions used for the perovskite alone, correspond to 1613 µmol g^−1^ h^−1^ and 963 µmol g^−1^ h^−1^ for 5 wt% and 15 wt% MHPs, respectively, showing a significant improvement of the photoactivity with respect to PEA_2_SnBr_4_ but also to g-C_3_N_4_ alone (81 µmol g^−1^ h^−1^) [16].

### 3.3. 0D Bi-Based and Sb-Based MHPs

In recent times, the substitution of the typical divalent metal cation of Pb with trivalent ones of Bi or Sb in has taken over the field of MHPs photovoltaic owing to the lead-like inactive outer shell s-orbital electrons of the latter [57]. Specifically, Bi^3+^ is isoelectronic to Pb^2+^, adopts a similar valence shell electron lone pair 6s^2^ and has a nearly equivalent effective ionic radius to Pb^2+^ [30,57,58]. However, as previously mentioned these trivalent cations are not able to form a continuous 3D structure and they lead to the formation of perovskites derivatives with low dimensionality (i.e., 2D or 0D) characterized by a very significant structural flexibility [44,55]. These MHPs derivatives can possess features not suitable for photovoltaic exploitation, such as indirect band gaps but very interesting for the photocatalytic production of H_2_ in aqueous solutions (e.g., long living excited states) [14,57,58].

A Bi-based 0D perovskite for H_2_ evolution was synthesized by Zhao et al. [31] by means of a solvothermal method. The obtained DMA_3_BiI_6_ single crystals showed a smooth surface as a rod-like shape with a length of 2 mm and a structure that belongs to the rhombohedral system. Particularly, the building units of DMA_3_BiI_6_ are isolated [BiI_6_]^3−^ octahedra surrounded by three DMA^+^ cations. The onset of the band gap is approximately 2.3 eV, considering that DMA_3_BiI_6_ is an indirect gap semiconductor and the fluorescence lifetime is 1.94 ns that is suitable for photocatalytic applications [75]. The photocatalytic measurements were performed by dispersing the perovskite powder in an aqueous HI-H_3_PO_2_ solution (1:4 volume ratio), connected to a vacuum system before the irradiation to exclude the oxygen, irradiating with a commercial light emitting diode and measuring the produced H_2_ after 6 h with a gas chromatograph equipped with a GC-TCD [31]. The formation of a transparent stable solution without any precipitation was observed also after 200 days, the stability of the perovskite in it was assessed by XRD analyses. Under irradiation at 425 nm with a light intensity of 8 mW, the highest H_2_ evolution rate (3.5 µmol g^−1^ h^−1^) was measured with a catalyst amount of 500 mg (i.e., 20 mg/mL). The photocatalytic performance was enhanced by photodeposition of Pt (0–1 wt%) on DMA_3_BiI_6_ crystals. Particularly, results showed the highest H_2_ evolution rate (46 µmol g^−1^ h^−1^) related to the MHP loaded with 1 wt% of Pt and a photocatalytic activity up to 100 h. The photocatalytic mechanism of this system can be explained considering the reactions in the HI-H_3_PO_2_ solution [31]. As reported in Figure 6a, the HI molecules are surrounded by the H_2_PO_2_^−^ ions to form colloidal particles, after the introduction of the perovskite crystals the DMA^+^ ions react with the exposed H_2_PO_2_^−^ ions to form another charged layer (Step 1) and Pt ions (i.e., Pt^x+^) are anchored on the surface of the colloids thanks to weak bond (Step 2). The inorganic [BiI_6_]^3−^ ions harvest the incident visible light, consequently the formation of electrons and holes is induced. Subsequently, these electrons may transfer to the Pt^x+^ and then react with H^+^ to generate H_2_ (Figure 6b). Therefore, the absorption properties of this photocatalytic system are dominated by the inorganic framework of the MHP, since in it the organic cation mainly plays a structural role [31,44,76]. In addition, it is important to evidence that the isolated characteristic of this well-dispersed system can inhibit the back reactions and enhance the charge separation important to have an efficient photocatalytic process [31,77].

The DMA_3_BiI_6_ perovskite was also employed by Tang et al. [78] to carry out the in-situ generation of heterojunctions with another Bi-based MHP, namely MA_3_Bi_2_I_9_, enhancing the photoinduced charge separation as well as the photocatalytic activity for H_2_ generation in aqueous HI solution. A solvothermal technique was used for the synthesis of the powered composite by adding different volume percentages (0%–10%) of dimethylformamide (DMF) in an isopropanol (IPA) solution. Experimental results showed that without the addition of DMF (BBP-0) the precursors originate the 0D MA_3_Bi_2_I_9_ with a sheet-like morphology and the crystal structure comprising two isolated face-sharing Bi_2_I_9_^3−^ octahedra. By using a DMF volume percentage of 10% (BBP-10) DMA_3_BiI_6_ perovskite is created with single rod-like morphology and the crystal structure containing isolated BiI_6_^3−^ octahedra. While by employing a volume percentage of 1% (BBP-1) or 5% (BBP-5) MA_3_Bi_2_I_9_/DMA_3_BiI_6_ heterojunctions are obtained with morphologies and XRD patterns resembling the ones of BBP-0 and BBP-10, respectively. All samples exhibited visible-light absorption in the range 300–650 nm with an indirect band gap (Figure 7a) and the positions of the VBM as of the CBM suitable for the visible-light photoinduced generation of H_2_ in an aqueous HI solution [9,14,78]. As schematically illustrated in Figure 7b, a driving force for the separation of photogenerated charge carriers is provided by the band alignment of the MA_3_Bi_2_I_9_ and DMA_3_BiI_6_ in the heterojunction. Electrochemical impedance spectroscopy (EIS) measurements, performed on a photoelectrode prepared by depositing the perovskite on FTO glass with the blade-coating method and photoluminescence (PL) investigations demonstrated the improvement of the charge separation and transfer between the heterojunction interface as well as a reduction of the nonradiative recombination [78]. The photocatalytic H_2_ production tests were performed by immersing the photocatalysts in an aqueous saturated HI solution maintained at 15 °C, irradiating the system with a 300 W Xe lamp (λ ≥ 420 nm) for 10 h and measuring the generated H_2_ with a gas chromatograph. Results confirmed that the Bi-based MHPs allow the HI splitting and the consequent H_2_ production. In particular, the BBP-5 composite shows the highest H_2_ production rate (198 µmol g^−1^ h^−1^), 15- and 8-fold increased with respect BBP-0 and BBP-10, respectively, which is stable 100 h under illumination [78]. Therefore the formation of the MA_3_Bi_2_I_9_/DMA_3_BiI_6_ heterojunction provides interesting insights into the H_2_ technology.

The possible application of the MA_3_Bi_2_I_9_ MHP alone in this field of the research was investigated by Guo et al. [29]. They synthesized the Bi-based material with a simple hydrothermal route. XRD patterns indicated that its structure agrees with the hexagonal phase and it remains stable in aqueous HI solution at different concentrations (i.e., different HI and water volume ratio). This result is relevantly different from that obtained for a MAPbI_3_ perovskite which is less stable in aqueous HI solution, creating a hydrated phase at low concentration of it [11]. This phenomenon could be related to the oxidation state of Bi^3+^, which is more stable than Pb^2+^, as to relativistic effects occurring in compounds containing heavy metals that limit their sensitivity to water [29,79]. The synthesized MA_3_Bi_2_I_9_ powder [29] showed a lamellar structure, with a smooth surface, of a bright red color with high absorbance in the visible light. Particularly, in accordance with the study previously mentioned [78], an E_g_ of about 2.0 eV was measured with the VBM as CBM values suitable for H_2_ reduction and iodine oxidation. The photocatalytic H_2_ evolution experiments were carried out by dispersing the lead-free perovskite in an aqueous saturated HI solution to which H_3_PO_2_ was added as reduction agent for I_3_^−^, irradiating with a 300 W Xe lamp (λ ≥ 400 nm) at 15 °C and quantifying the evolved H_2_ after 10 h with gas chromatography [29]. The obtained results showed a H_2_ evolution of 13 µmol g^−1^ h^−1^ confirming that H_2_ is a product of the MA_3_Bi_2_I_9_ powder in aqueous HI solution. Specifically, under visible-light irradiation, the photogenerated electrons in the perovskite are excited into the CB, separated from the photogenerated holes, where H^+^ of dissociated HI is reduced to H_2_ by these electrons. Meanwhile, the photogenerated holes in the VB are used to oxidize I^−^ to I_3_^−^ which is reduced from H_3_PO_2_ to achieve stable long-term reactions [29]. To optimize the efficiency of the system, photoreduction was utilized to deposit Pt on the perovskite surface. A H_2_ evolution rate of about 169 µmol g^−1^ h^−1^, 14 times enhanced than that of the as synthesized MHP, was determined. This rate showed no significant decrease after 70 h of repeated experiments.

For the H_2_ evolution in aqueous HI solutions also Pb-free totally inorganic 0D MHPs were investigated by Chen et al. [42]. Particularly, they prepared Cs_3_Bi_2_I_9_, Cs_3_Sb_2_I_9_ (and Cs_3_Bi_2x_Sb_2-2x_I_9_ (x equals to 0.1, 0.3, 0.5, 0.7 or 0.9) powder samples, with particles of irregular shapes (5–10 µm in size), using a co-precipitation method. The XRD patterns revealed that they have a hexagonal structure, even if for Cs_3_Bi_2x_Sb_2-2x_I_9_ the diffraction peaks shift to higher 2θ values because the Sb^3+^ ion is smaller than the Bi^3+^ one [42]. The UV-vis diffuse reflectance spectra showed that Cs_3_Bi_2x_Sb_2-2x_I_9_ samples possess a smaller band gap than Cs_3_Bi_2_I_9_ and Cs_3_Sb_2_I_9_ (1.9 eV) with the smallest value (1.6 eV) related to Cs_3_Bi_2x_Sb_2-2x_I_9_ with x = 0.3 (i.e., Cs_3_Bi_0.6_Sb_1.4_I_9_) which exhibits a high absorption in the visible light region. Preliminary H_2_ evolution experiments performed on Cs_3_Bi_2_I_9_, in aqueous saturated HI solution containing H_3_PO_2_ (9:1 volume ratio), at 15 °C, under visible light irradiation (300 W Xe lamp with an AM 1.5 cut-off filter) for 10 h, showed a very low efficiency (2 µmol h^−1^ g^−1^) due to the presence of Bi^3+^ ions. Indeed, the dissolution of the Bi-based material in solution creates (Bi_2_I_9_)^3−^ anions which produce Bi^3+^ cations. These latter capture electrons, generated by the absorption of light, more readily than H^+^ protons creating elemental Bi as well as hindering the H_2_ production [42,80]. To reduce this phenomenon, Cs_2_CO_3_ was added to the employed solution. In fact, due to the reaction of the (Bi_2_I_9_)^3−^ anions with the added Cs^+^ ions, a decrease of the formed Bi^3+^ was obtained and a higher H_2_ evolution (22 µmol h^−1^ g^−1^) was measured by gas chromatography [42]. The H_2_ evolution tests carried out for Cs_3_Bi_2x_Sb_2-2x_I_9_ samples showed that they exhibit a higher photocatalytic activity than Cs_3_Bi_2_I_9_ but also than Cs_3_Sb_2_I_9_ (about 60 µmol h^−1^ g^−1^) with the maximum for Cs_3_Bi_0.6_Sb_1.4_I_9_ (157 µmol h^−1^ g^−1^) without significant decline after 50 h of activity. Under the same photocatalytic reaction conditions with irradiation light through a band-pass filter of 420 nm and a masked area of π cm^2^ (light irradiance at 100 mW cm^−2^), the AQE was also calculated. For Cs_3_Bi_0.6_Sb_1.4_I_9_ an AQE of 1.21% was determined. To further improve the HI splitting performance of Cs_3_Bi_0.6_Sb_1.4_I_9_, Pt was loaded using the photoreduction method. A H_2_ evolution rate of 926 µmol h^−1^ g^−1^ and an AQE of 0.319% were so determined. In order to better explain the measured catalytic performances of samples, their photoelectrochemical properties were examined by using thin films produced by spin-coating in dichloromethane containing 0.1 M tetrabutylammonium hexafluorophosphate (TBAPF6). The photocurrents of Cs_3_Bi_2_I_9_, Cs_3_Sb_2_I_9_ and Cs_3_Bi_0.6_Sb_1.4_I_9_ films, measured at 0 V versus Ag/AgCl electrode, indicated that reduction reactions take place on their surfaces and that much more photocarriers are accumulated on the surface of Cs_3_Bi_0.6_Sb_1.4_I_9_ than on that of Cs_3_Bi_2_I_9_ as Cs_3_Sb_2_I_9_ films. In addition, Cs_3_Bi_2x_Sb_2-2x_I_9_ showed stronger PL intensity than Cs_3_Bi_2_I_9_ or Cs_3_Sb_2_I_9_ indicating that nonradiative recombination is suppressed in these MHPs, as confirmed by surface photovoltage measurements. The PL decay times revealed that Cs_3_Bi_2x_Sb_2-2x_I_9_ has a more rapid charge transfer as a more efficient electron-hole separation than Cs_3_Bi_2_I_9_ or Cs_3_Sb_2_I_9_ which is beneficial for the photocatalytic activity [42].

### 3.4. Double MHPs

Double perovskites have been synthesized since the 1970s, however, only a few types of them have been applied so far in solar cells [57,60,62]. In particular, the most popular double perovskite used in PV applications is the inorganic Pb-free Cs_2_AgBiBr_6_ which is characterized by properties similar to those of the typical MAPbI_3_ and MAPbBr_3_ 3D MHP (e.g., high defect tolerance) [57]. It has an indirect band gap of about 2.0 eV and good moisture as well as thermal stability. Its structural network is ideally composed of corner shared octahedra, alternately formed by the Ag^+^ and Bi^3+^ metal cations with the Br^−^ anions and inorganic Cs^+^ ions that occupy the generated octahedral cavities [60,61,62]. Due to the good stability as to the optoelectronic properties, nowadays Cs_2_AgBiBr_6_ perovskite has been also investigated for the photocatalytic H_2_ generation from aqueous solutions [30,32].

Shi et al. [30] synthesized Cs_2_AgBiBr_6_ powder by using the method of supersaturated precipitation in HBr solution containing an excess of precursors. The exact crystal structure of the realized perovskite was resolved by XRD analyses and confirmed the formation of a typical cubic 3D structure with Ag^+^ and Bi^3+^ cations that alternately occupy the centers of octahedra formed with Br^−^ anions. In these octahedra the lengths of Ag-Br and Bi-Br bonds are comparable due to the larger ionic radius of Ag^+^ than Bi^3+^ [30,81]. The powder [30] appeared orange presenting an absorption onset at 590 nm with the attended indirect band gap of 2.0 eV. Additionally, XRD patterns and UV-vis spectra demonstrated an excellent stability in air as well as under light exposure up to 28 days. Ultraviolet photoelectron spectroscopy (UPS) characterization demonstrated that the atomic orbitals of Ag participate in the band structure composition of the obtained material. The binding energy of electrons around the different elements, evaluated with XPS, showed a well dispersed electronic distributions indicating good carrier transport. To obtain information on the photoelectric properties of Cs_2_AgBiBr_6_, various factors were calculated by using a three-electrode configuration workstation, containing an electrolyte solution of dichloromethane containing 0.1 M TBAPF6, characterized by perovskite deposited on FTO glass by spin-coating, a Pt plate and Ag/Ag^+^ (0.01M AgNO_3_ in acetonitrile) as counter, reference and working electrodes, respectively. An exciton binding energy of 85 meV, an electron mobility of 120 m^2^ V^−1^ s^−1^, a hole mobility of 3.6 m^2^ V^−1^ s^−1^, an electron relaxation time of 2.3 × 10^5^ fs and a hole relaxation time of 4.8 × 10^3^ fs were specifically determined. The PL peak of the Cs_2_AgBiBr_6_ exhibited an apparent red-shift relative to its optical band gap probably originated from emission of defect states [30,82]. The average PL lifetime of 73 ns [30], is favorable for photogenerated charge separation and transfer to be involved in subsequent surface reactions. The photocatalytic H_2_ evolution tests were carried out by dispersing the perovskite powder in aqueous saturated HBr or HBr-H_3_PO_2_ (5:1 volume ratio) solutions, irradiating with a 300 W Xe lamp (λ ≥ 420 nm) for 3 h, at 15 °C. Experimental results showed a photocatalytic H_2_ evolution activity of about 0.7 µmol g^−1^ h^−1^ and 18 µmol g^−1^ h^−1^ in the HBr solution and HBr-H_3_PO_2_, respectively [30]. Moreover, the perovskite exhibited an excellent photocatalytic stability throughout several cycles (13 h) in terms of H_2_ production in both HBr and HBr-H_3_PO_2_ solutions. As attended, an enhanced H_2_ evolution rate of about 22 µmol g^−1^ h^−1^ and 55 µmol g^−1^ h^−1^ in the HBr and HBr-H_3_PO_2_ solution, respectively, was measured by preparing Pt-loaded Cs_2_AgBiBr_6_.

Wang et al. [32] combined the same Pb-free double perovskite with reduced graphene oxide (RGO), an amorphous material with good electron transfer properties and stability to acids, to obtain an efficient composite for H_2_ evolution in saturated aqueous HBr solution. Specifically, the authors synthesized Cs_2_AgBiBr_6_ from aqueous solution and prepared Cs_2_AgBiBr_6_/xRGO (x = 1%, 2.5% or 5% mass ratio) via photoreduction. The XRD patterns showed the high crystalline cubic double perovskite structure of Cs_2_AgBiBr_6_ that is maintained in Cs_2_AgBiBr_6_/xRGO composites. Cs_2_AgBiBr_6_ particles showed an irregular bulk shape in the range of several micrometers, while in composites winkled RGO and staked perovskite grains were observed. The double MHP displayed an absorption onset at about 750 nm and the absorption increased sharply between 500 and 600 nm because of its indirect bandgap nature. With the raise RGO loading, the composite samples look appeared darkened with more visible light absorption than pure orange perovskite, which is beneficial for visible light photocatalytic H_2_ evolution. The photocatalytic activity of synthesized materials was evaluated by immersing them in aqueous saturated HBr and H_3_PO_2_ solutions (5:1 volume ratio) under visible light irradiation by a 300 W Xe lamp (λ ≥ 420 nm) for 10 h at 5 °C. Preliminary results showed that Cs_2_AgBiBr_6_ produced 0.6 µmol g^−1^ h^−1^ of H_2_ indicating a poor activity that was increased at 1 µmol g^−1^ h^−1^ after the photodeposition of Pt [32]. The H_2_ production was significantly increased by using Cs_2_AgBiBr_6_/xRGO composites and the highest value (49 µmol g^−1^ h^−1^) was measured for Cs_2_AgBiBr_6_/2.5%RGO. Furthermore, the production of H_2_ by employing 2.5% RGO alone was only 4 µmol g^−1^ h^−1^, demonstrating the synergistic effect between the double perovskite and the graphene oxide. The effect of H_3_PO_2_ concentration in the saturated HBr solution containing Cs_2_AgBiBr_6_/2.5% RGO was also investigated by repeating the H_2_ evolution experiments with the same experimental conditions but varying their volume ratio (i.e., HBr:H_3_PO_2_ = 1.6:1, 3:1, 5:1 or 7:1). The best photocatalytic performance was obtained for 5:1 ratio. In this case an AQE of 0.16% was determined, under 450 nm incident light irradiation. The photocatalytic stability of Cs_2_AgBiBr_6_/2.5% RGO was studied and no significant decrease in the H_2_ evolution was observed after 120 h of photoreaction demonstrating the composite stability in strong acid solution. In order to clarify the great improvement on H_2_ evolution attributable to the preparation of perovskite/RGO composites, PL analyses and photoelectrochemical measurements on a working electrode obtained by the deposition of catalysts on FTO glass located in a workstation containing Ag/AgCl 3M as reference electrode and Pt mesh as counter electrode in an electrolyte solution of dichloromethane containing 0.1 M TBAPF6 were performed. Results allowed to conclude that for Cs_2_AgBiBr_6_/xRGO samples, a rapid transfer of photogenerated electron from the perovskite to the RGO is verified, suppressing the charge recombination and leading to the improvement of the H_2_ production [32,83,84]. The Cs_2_AgBiBr_6_/2.5% RGO sample exhibited the highest photocurrent as the lowest charge-transfer resistance than other samples, corresponding to the best photocatalytic H_2_ evolution activity [32]. Therefore, based on the obtained results, the mechanism schematized in Figure 8 was proposed. In the saturated acid solution, where the Cs_2_AgBiBr_6_ experience the dynamic precipitation-dissolution equilibrium process, RGO acts as an attachment for perovskite particles. Under visible light irradiation, the latter generate electrons and holes. The generated electrons transfer to conductive RGO and reduce H^+^ to produce H_2_. Meanwhile, Br^−^ is oxidized to Br_3_^−^ by the holes on Cs_2_AgBiBr_6_ particles. Thus, continuous photocatalytic H_2_ evolution from HBr splitting occurs.

**Table 1 nanomaterials-11-00433-t001:** Summary of the photocatalytic metal halide perovskite (MHP) systems discussed in this review.

Photocatalyst	Aqueous Media	Measurement Conditions	Activity (μmol g^−1^ h^−1^)	Ref.
DMASnBr_3_/Pt 10 wt%	DI water + 10% *v*/*v* TEOA	simulated solar light 500 Wm^−2^ for 5 h UV filter	111	[34]
g-C_3_N_4_/DMASnBr_3_ 33 wt%	DI water + 10% *v*/*v* TEOA	simulated solar light 500 Wm^−2^ for 6 h UV filter	19	[17]
g-C_3_N_4_/DMASnBr_3_ 33 wt%/Pt 3 wt%	DI water + 10% *v*/*v* TEOA	simulated solar light 500 Wm^−2^ for 6 h UV filter	1730	[17]
g-C_3_N_4_/DMASnBr_3_ 33 wt%/Pt 3 wt%	DI water + glucose 0.1M	simulated solar light 500 Wm^−2^ for 6 h UV filter	300	[17]
DMASnI_3_	DI water	300 W Xe lamp for 5 h a 10 °C	3	[15]
PEA_2_SnBr_4_/Pt 3 wt%	DI water + 10% *v*/*v* TEOA	simulated solar light 500 Wm^−2^ for 6 h UV filter	4	[16]
g-C_3_N_4_/PEA_2_SnBr_4_ 5 wt%/Pt 3 wt%	DI water + 10% *v*/*v* TEOA	simulated solar light 500 W m^−2^ for 6 h	1613	[16]
DMA_3_BiI_6_	20 mg/mL in aqueous HI:H_3_PO_2_ 1:4	commercial LED for 6 h 425 nm, light intensity 8 mW	3.5	[31]
DMA_3_BiI_6_/Pb 1wt%	20 mg/mL in aqueous HI:H_3_PO_2_ 1:4	commercial LED for 6 h 425 nm, light intensity 8 mW	46	[31]
DMA_3_BiI_6_	aqueous saturated HI solution at 15 °C	300 W Xe lamp (λ ≥ 420 nm) for 10 h	25	[78]
DMA_3_BiI_6_/MA_3_Bi_2_I_9_	aqueous saturated HI solution at 15 °C	300 W Xe lamp (λ ≥ 420 nm) for 10h	198	[78]
MA_3_Bi_2_I_9_	aqueous saturated HI solution + H_3_PO_2_	300 W Xe lamp (λ ≥ 420nm) for 10 h	13	[29]
MA_3_Bi_2_I_9_/Pt	aqueous saturated HI solution + H_3_PO_2_	300 W Xe lamp (λ ≥ 420nm) for 10 h	169	[29]
Cs_3_Bi_2_I_9_	aqueous saturated HI + H_3_PO_2_ 9:1	300 W Xe lamp for 10 h at 15 °C AM 1.5 cut-off filter	2	[42]
Cs_3_Bi_2_I_9_	aqueous saturated HI + H_3_PO_2_ 9:1 + Cs_2_CO_3_)	300 W Xe lamp for 10 h at 15 °C AM 1.5 cut-off filter	22	[42]
Cs_3_Sb_2_I_9_	aqueous saturated HI + H_3_PO_2_ 9:1 + Cs_2_CO_3_	300 W Xe lamp for 10 h at 15 °C AM 1.5 cut-off filter	60	[42]
Cs_3_Bi_0.6_Sb_1.4_I_9_	aqueous saturated HI + H_3_PO_2_ 9:1 + Cs_2_CO_3_	300 W Xe lamp for 10h at 15 °C AM 1.5 cut-off filter	157	[42]
Cs_3_Bi_0.6_Sb_1.4_I_9_/Pt	aqueous saturated HI + H_3_PO_2_ 9:1 + Cs_2_CO_3_	300W Xe lamp for 10 h at 15°C AM 1.5 cut-off filter	926	[42]
Cs_2_AgBiBr_6_	aqueous saturated HBr solution	300 W Xe lamp for 3 h (λ ≥ 420 nm) at 15 °C	0.7	[30]
Cs_2_AgBiBr_6_	aqueous saturated HBr:H_3_PO_2_ 5:1 solution	300 W Xe lamp for 3 h (λ ≥ 420 nm) at 15 °C	18	[30]
Cs_2_AgBiBr_6_/Pt	aqueous saturated HBr:H_3_PO_2_ 5:1 solution	300 W Xe lamp for 3 h (λ ≥ 420 nm) at 15 °C	22	[30]
Cs_2_AgBiBr_6_/Pt	aqueous saturated HBr:H_3_PO_2_ 5:1 solution	300 W Xe lamp for 3 h (λ ≥ 420 nm) at 15 °C	55	[30]
Cs_2_AgBiBr_6_	aqueous saturated HBr + H_3_PO_2_ solution 5:1	300 W Xe lamp for 10 h (λ ≥ 420 nm) at 5 °C	0.6	[32]
Cs_2_AgBiBr_6_/Pt	aqueous saturated HBr + H_3_PO_2_ solution 5:1	300 W Xe lamp for 10 h (λ ≥ 420 nm) at 5 °C	1	[32]
Cs_2_AgBiBr_6_/2.5 wt% RGO	aqueous saturated HBr + H_3_PO_2_ solution 5:1	300 W Xe lamp for 10 h (λ ≥ 420nm) at 5 °C	49	[32]

## 4. Conclusions and Perspectives

In the last decade MHPs have brought an unprecedent revolution to the field of photovoltaics, reason why many efforts within the scientific community have been devoted toward the maximization of their potential for this application, mainly targeting the obtainment of compact and uniform thick polycrystalline films characterized by excellent and anisotropic charge transport and long charge diffusion length. The ideal candidates for this purpose are Pb-based perovskites, particularly MAPbI_3_ and closely related materials. However, the potential of halide perovskites is emerging for alternative applications as well. Among them the solar-driven H_2_ production employing perovskites as photocatalyst is one of the most interesting and unexplored. In this paper we reviewed some of the seminal perovskite structures that were able to sustain the H_2_ evolution in aqueous solutions under solar irradiation. We specifically focused on Pb-free perovskites, thus on materials that would be sustainable in terms of environmental impact.

In this contest the high structural and compositional variability represents a powerful tool aiming to reach material stability in water and consequently at high photocatalytic activity towards H_2_ production, with much less constrains with respect to the material exploitation for PV, opening almost endless possibilities. In fact the suitability to work as photocatalyst is not strictly contingent upon the charge transport within the layer nor the obtainment of a uniform film, allowing the exploitation of many more metal (e.g., Bi, Sn, Ge) cation combinations replacing Pb^2+^, of Cs^+^, Ag^+^, DMA^+^ and so forth, replacing MA^+^ and of different halides (I^−^, Br^−^, Cl^−^) within the structure, also leading to low dimensional structures different synthetic/deposition methods. We underlined the importance, foreseeing a practical and scalable application of this materials for H_2_ evolution, of developing intrinsically stable MHPs to overcome the dynamic dissolution/precipitation equilibrium requiring strongly acidic conditions, clearly source of problems such as corrosion of reactors, safety and so on. In fact, to date the majority of perovskite-based photocatalysts have been employed in HI or HBr solutions, while only a few perovskite systems have been reported to possess an intrinsic stability in water, DMASnBr_3_ [17,34], DMASnI_3_ [15] and PEA_2_SnBr_4_ [16], which are extremely promising for future developments. It seems that the hydrophilic ammonium cation of MA^+^ may be the cause for water instability issues, by triggering the water ingress in the perovskite structure. Based on this hypothesis, the change of the monovalent cation with one less prone to interact with water, such as Cs^+^, FA^+^ or DMA^+^ or better with one water repulsive such as PEA^+^, could importantly impact on the interaction with water, thus on the material stability, in such environment. However, it is necessary to deepen the study of these materials to fully understand the reasons behind these characteristics and, consequently, to develop a wide range of perovskite materials with these same features. Additional key advantages would come from the development of intrinsically water-stable MHPs and derivatives. Among them (i) the possible exploration of different pH conditions, (ii) the combination of MHPs with co-catalysts materials with the aim of compartmentalize the catalytic processes improving the efficiencies, (iii) the possibility of depositing MHPs with various technique aiming at superior exposed surface, thin porous film thus photoelectrode preparation can be distinguished.

It is worth noticing that at this seminal stage of research, none of the developed water-stable Pb-free materials have been characterized in film but only as powder dispersed in water. Thus, besides the obtainment of intrinsically water-stable materials, a necessary next step toward the development of this technology is to translate the perovskite powders into polycrystalline film supported on conductive transparent substrates (e.g., ITO or FTO) as part of a PEC cell. As mentioned above, a wide range of techniques can be employed, from sputtering to doctor blading, powder pressure, single step coating, since the film flatness here is rather an unwanted complication hindering the obtainment of extended contact surface between the photocatalyst and the reaction aqueous media. Those approaches would allow more degree of freedom in engineering the material deposition. Perovskites are also well known for the high compatibility with different material, for example for their capability of growing upon different substrates, therefore as discussed within the review it is possible to combine them with partner co-catalysts (e.g., g-C_3_N_4_, TiO_2_, Fe_2_O_3_, etc.), in different layouts, for example meso- and/or nanostructured composites thus further improve the H_2_ evolution efficiency.

The very preliminary results discussed in this review and the research directions foreseen pave the way to the realization of a completely new class of photocatalysts with the potential to bring a significant contribution to the field of solar to fuel, supporting the transition towards communities grounded on eco-sustainable carbon-neutral energy supplies.

## Figures and Tables

**Figure 1 nanomaterials-11-00433-f001:**
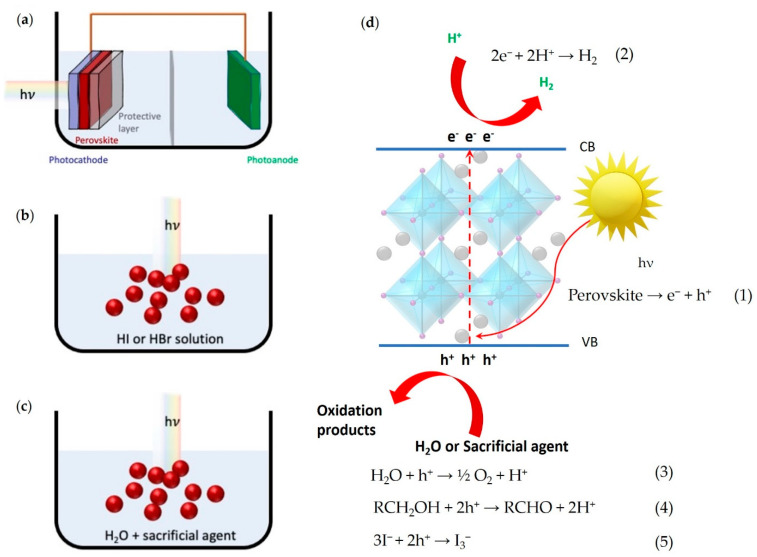
Solar-driven perovskite-based H_2_ production systems: (**a**) photoelectrochemical (PEC) cell; (**b**) particulate photocatalyst system in dynamic equilibrium; (**c**) particulate water-stable photocatalyst system. (**d**) Schematic representation of the processes on the perovskite photocatalyst surface under irradiation and possible reactions involved in the different systems (Equations (1)–(5)).

**Figure 2 nanomaterials-11-00433-f002:**
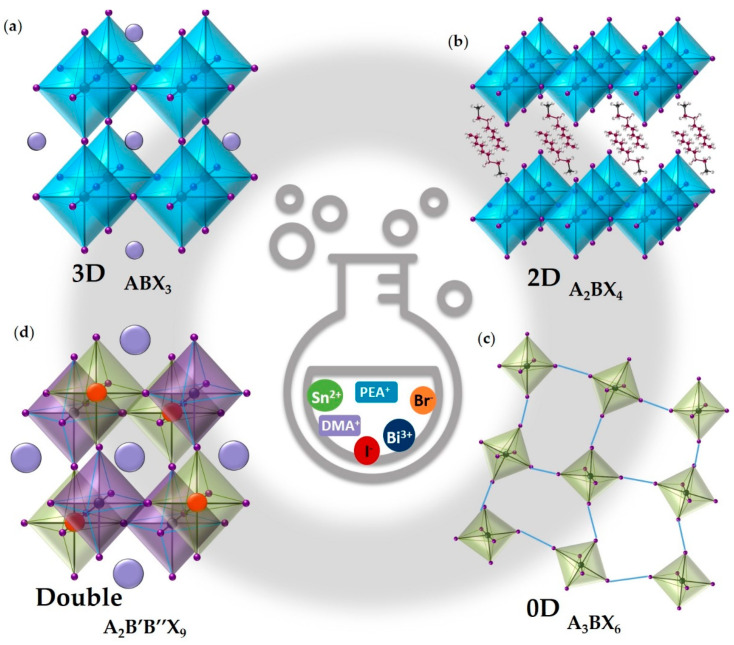
Examples of schematic crystal structures with the related chemical formula of (**a**) 3D, (**b**) 2D, (**c**) 0D and (**d**) double MHPs used for H_2_ evolution from aqueous solutions.

**Figure 3 nanomaterials-11-00433-f003:**
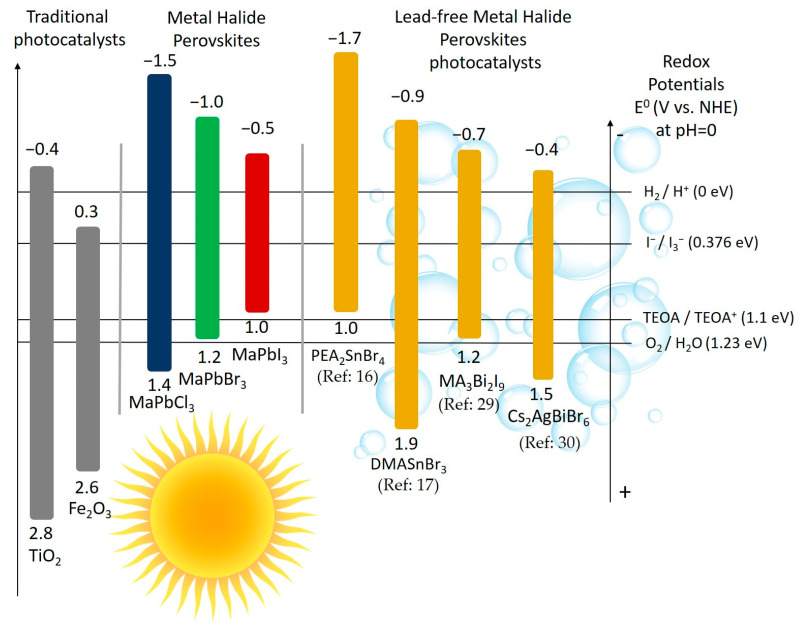
Redox potentials of TiO_2_ and Fe_2_O_3_ taken as example of traditional photocatalysts, Pb-based halide perovskites and Pb-free MHPs already exploited for hydrogen evolution processes.

**Figure 4 nanomaterials-11-00433-f004:**
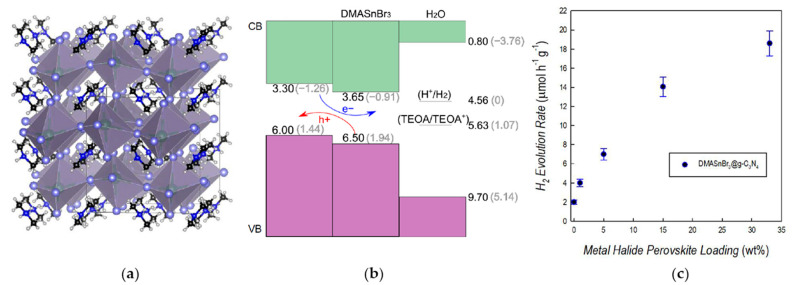
(**a**) Schematic representation of DMASnBr_3_ crystal structure, (**b**) valence band (VB) to conduction band (CB) edges of g-C_3_N_4_ and DMASnBr_3_ aligned with the band edges of liquid water and with the H^+^/H_2_ and triethanolamine (TEOA)/TEOA^+^ redox level through the vacuum level. Values (eV) are referred to the vacuum level (black) and to the standard H_2_ electrode (SHE, grey), (**c**) H_2_ evolution rates for DMASnBr_3_@g-C_3_N_4_ composites at different percentages of MHP loading in 10% *v*/*v* TEOA aqueous solution. Reproduced from [17], with permission from Whiley, 2020.

**Figure 5 nanomaterials-11-00433-f005:**
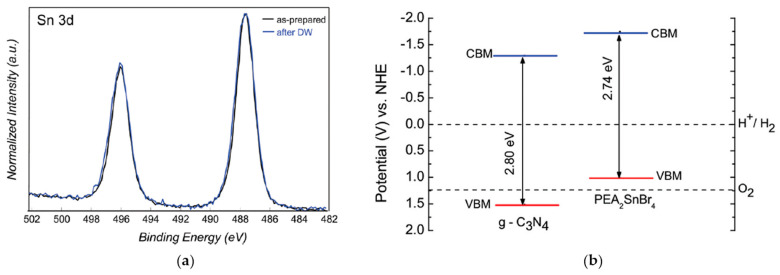
(**a**) Sn 3d high-resolution X-ray photoelectron spectroscopy (XPS) spectra of the as-prepared PEA_2_SnBr_4_ and after storage in water; (**b**) Calculated band edge positions (continuous lines) for g-C_3_N_4_ and PEA_2_SnBr_4_ relative to Normal Hydrogen Electrode (NHE) potential; the dashed lines indicate the water redox reaction potentials. Reproduced with permission [16] Copyright 2020, Royal Society of Chemistry.

**Figure 6 nanomaterials-11-00433-f006:**
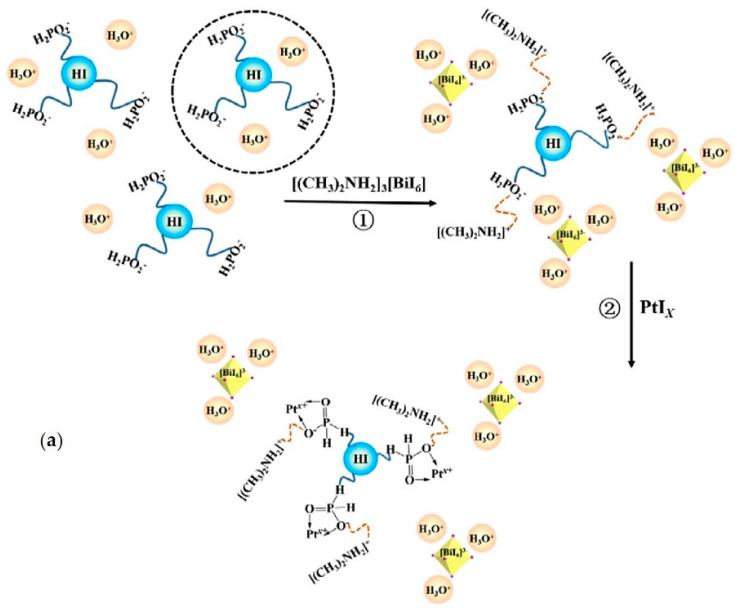
Schematic illustration of (**a**) the conversion process in the colloidal HI-H_3_PO_2_ aqueous solution and (**b**) the proposed mechanism for the H_2_ generation. [(CH_3_)_2_NH_2_]^+^ ions correspond to DMA^+^ ones. Reproduced from [31], with permission from Elsevier, 2018.

**Figure 7 nanomaterials-11-00433-f007:**
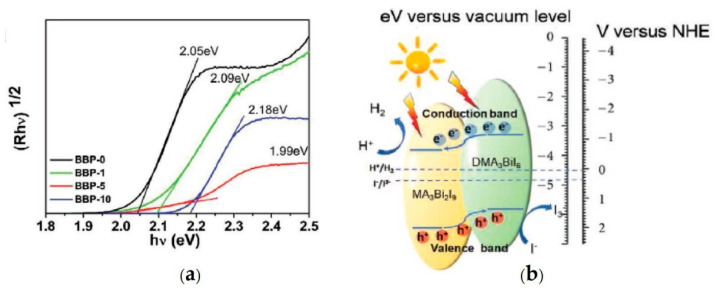
(**a**) Indirect band gap Tauc plot of as synthesized samples; (**b**) Energy level diagram of a MA_3_Bi_2_I_9_/DMA_3_BiI_6_ heterojunction for the photocatalytic HI splitting. Reproduced from [78], with permission from Wiley, 2020.

**Figure 8 nanomaterials-11-00433-f008:**
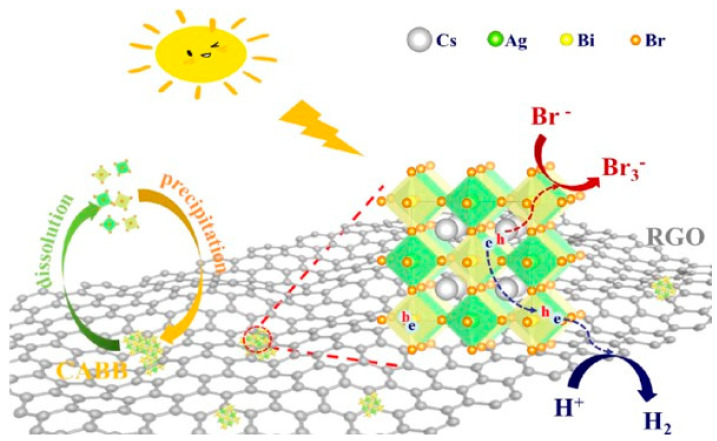
Schematic mechanism of H_2_ evolution in aqueous HBr solution by Cs_2_AgBiBr_6_ (CABB)/RGO composite under visible-light irradiation. Reproduced from [32], with permission from Elsevier, 2020.

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
