# Peer review of "Lead-Free Metal Halide Perovskites for Hydrogen Evolution from Aqueous Solutions"

_nanomaterials, 2021, doi:10.3390/nano11020433_

Round 1

Reviewer 1 Report

The minireview contains useful information on the photocatalytic properties of Pb-free metal halide perovskites (MHPs) for hydrogen production and it will certainly be of interest to readers of the journal Nanomaterials. The materials composition and crystalline structure impacting on the catalytic process of hydrogen evolution has been comprehensively analyzed. The paper is generally well written and can be accepted, however, two issues require clarification. The water stable Pb-free MHPs described in the paper have been developed (at present) as powder dispersed in water. For readers of the Nanomaterials journal, it will be interesting to know the morphology and dimension of the powders/nanoparticles/nanocrystals of the characteristic MHPs. In Figure 3, authors used only TiO2 and Fe2O3 as traditional photocatalysis. The choice of these materials for comparison with MHPs should be discussed. TiO2 is an inactive photocatalyst under visible light irradiation since it can be excited solely by the UV light. Non-metal doping, metal coupling, or fabrication of heterogeneous structures with narrow bandgap semiconductors (e.g., metal (di)chalcogenides) are used to enhance the photocatalytic activity. Other metal oxides have now been developed (for example, tungsten oxides and ternary metal oxides) and hybrids/heterostructures of metal oxides with metal chalcogenides (WS2, MoSx et al.) which exhibit high performance for the photo-activated hydrogen evolution in aqueous solutions. Some recent information on other nanomaterials for photo-activated hydrogen evolution can be found in the articles: Zhang Y. et al. Enhancing hydrogen evolution by photoelectrocatalysis of water splitting over a CdS flowers-loaded TiO2 nanotube array film on the Ti foil substrate. Ceramic International 46(2020) 17606; Zhang S. et al. Layered WS2/WO3 Z-scheme photocatalyst constructed via an in situ sulfurization of hydrous WO3 nanoplates for efficient H2 generation, Appl. Surf. Sci. 529 (2020) 147013; V. Fominski et al. Performance and Mechanism of Photoelectrocatalytic Activity of MoSx/WO3 Heterostructures Obtained by Reactive Pulsed Laser Deposition for Water Splitting, Nanomaterials 2020, 10 (5), 871.

I recommend the authors to check the text and correct the typos. For example:

Line 162 – Figure 2a ? (Figure 1a)

Line 445 – H2 ? (H2)

Line 876-878 – Ref.64?

Author Response

Manuscript ID: nanomaterials-1100078

Title: Lead-free metal halide perovskites for hydrogen evolution from aqueous solutions

We would like to thank all the reviewers for their appreciation to the work as well as to their careful evaluations/suggestions which allowed us to increase its soundness and potential impact.

The manuscript was accordingly amended, and revisions were reported in red.

Response to Reviewer 1

  • The water stable Pb-free MHPs described in the paper have been developed (at present) as powder dispersed in water. For readers of the Nanomaterials journal, it will be interesting to know the morphology and dimension of the powders/nanoparticles/nanocrystals of the characteristic MHPs.

We thank the reviewer for this fundamental question. We fully reviewed all the cited and discussed literature, and where possible we add the information related to the MHPs dispersion properties in terms of morphology and dimension. As the field is at the very beginning not in every work the authors report such details, we add a sentence to our discussion underlining how this should be done more often in the paper for the sake of comparison.

Sentence added at the end of section 2 (from line 200 to line 205):

“Other important information that have direct impact on the photocatalytic performance, and which are often missing in the reviewed works, are the one related to the morphology/dimension of the particulate. Those features should be ideally highlighted in future works due to their impact on the MHPs properties. On one hand, crystal dimension influence material optics, on the other hand morphology determines the surface exposed to the reaction thus the photocatalyst activity.”

  • In Figure 3, authors used only TiO2 and Fe2O3 as traditional photocatalysis. The choice of these materials for comparison with MHPs should be discussed. TiO2 is an inactive photocatalyst under visible light irradiation since it can be excited solely by the UV light. Non-metal doping, metal coupling, or fabrication of heterogeneous structures with narrow bandgap semiconductors (e.g., metal (di)chalcogenides) are used to enhance the photocatalytic activity. Other metal oxides have now been developed (for example, tungsten oxides and ternary metal oxides) and hybrids/heterostructures of metal oxides with metal chalcogenides (WS2, MoSx et al.) which exhibit high performance for the photo-activated hydrogen evolution in aqueous solutions. Some recent information on other nanomaterials for photo-activated hydrogen evolution can be found in the articles: Zhang Y. et al. Enhancing hydrogen evolution by photoelectrocatalysis of water splitting over a CdS flowers-loaded TiO2 nanotube array film on the Ti foil substrate. Ceramic International 46(2020) 17606; Zhang S. et al. Layered WS2/WO3 Z-scheme photocatalyst constructed via an in situ sulfurization of hydrous WO3 nanoplates for efficient H2 generation, Appl. Surf. Sci. 529 (2020) 147013; V. Fominski et al. Performance and Mechanism of Photoelectrocatalytic Activity of MoSx/WO3 Heterostructures Obtained by Reactive Pulsed Laser Deposition for Water Splitting, Nanomaterials 2020, 10 (5), 871.

Thanks again the reviewer for the stimulating question. The choice of the photocatalysts for comparison has been related to historical reasons: TiO2 and Fe2O3 are among the first example of material used for the purpose. Additionally, the current record for hydrogen evolution rates for aqueous solutions is still related to the use of TiO2 as photocatalyst and the figure reporting their level (Figure 3) has the aim to show the potential of  MHPs in terms of band gap tunability. However the examples cited by the review are of interest for a comparison with the current literature on the field, therefore we included those in the revised version  our work (References 63-66).

Revised text: “In the Figure some of the traditional photocatalysts, namely TiO2 and Fe2O3 are also reported[7,9]. Clearly, many other examples of efficient alternative photocatalytic materials could be found in literature, (e.g. metal dichalcogenides and oxides [63] as well as hybrid structures [64–66]) however the discussion of those alternatives lies outside the focus of this minireview. Figure 3 highlights the excellent tunability of Pb-free MHPs properties, in particular the water reduction ability related to the relative position of their CB; typically, negative enough for H2 generation.”

  • I recommend the authors to check the text and correct the typos. For example:

Line 162 – Figure 2a ? (Figure 1a)

Line 445 – H2 ? (H2)

Line 876-878 – Ref.64?

We corrected those points in the revised version of the manuscript.

Reviewer 2 Report

This is an interesting study on perovskites, which is a topic of significant interest especially in areas as diverse as technology, and clean energy. The authors of the study have carefully carried discussed the basic science and the work is equipped with adequate tables and figures. The conclusion is also perfect. I recommend publication of the work in its current form.

Author Response

Manuscript ID: nanomaterials-1100078

Title: Lead-free metal halide perovskites for hydrogen evolution from aqueous solutions

We would like to thank all the reviewers for their appreciation to the work as well as to their careful evaluations/suggestions which allowed us to increase its soundness and potential impact.

The manuscript was accordingly amended, and revisions were reported in red.

Response to Reviewer 2

We thank again the reviewer for the very positive comments on our work.

Reviewer 3 Report

In this paper, the authors provided a minireview on the recent advancements in the use of lead-free metal halide perovskites (MHPs) structures stable in aqueous solutions for efficient solar-driven perovskite-based hydrogen production.

The paper is properly divided in sections and sub-sections, but it needs some corrections before its publication on the journal.

  • The authors should improve the language in the text, in order to eliminate some typing errors and to make some statements more understandable (for example the first statement in the abstract);
  • Due to the nature of this paper (a minireview) the authors should extend the literature survey, by adding more recent papers in this field: some examples are 10.1007/s00706-017-1933-9, 10.1039/C9SC00453J, 10.1016/j.nanoen.2020.105526;
  • The authors at line referred to figure 2a: perhaps they meant figure 1a;
  • The authors used multiple references in the text: they should try to evidence the contribution of the single reference to the discussion (for example at line 164);
  • The authors should express the measuring units more homogeneously, by using always either the apexes mode or the “/” mode (as at line 175);
  • Perhaps in equation 6 a conversion factor is missing for having a uniform measuring unit;
  • The sections 3.3 and 3.4 have the same title: is there an error?
  • Due to the high number of symbols and acronyms, perhaps a section with their lists could be useful;

Author Response

Manuscript ID: nanomaterials-1100078

Title: Lead-free metal halide perovskites for hydrogen evolution from aqueous solutions

We would like to thank all the reviewers for their appreciation to the work as well as to their careful evaluations/suggestions which allowed us to increase its soundness and potential impact.

The manuscript was accordingly amended, and revisions were reported in red.

Response to Reviewer 3

  • The authors should improve the language in the text, in order to eliminate some typing errors and to make some statements more understandable (for example the first statement in the abstract);

We tried to improve the language throughout our manuscript.

  • Due to the nature of this paper (a minireview) the authors should extend the literature survey, by adding more recent papers in this field: some examples are 10.1007/s00706-017-1933-9, 10.1039/C9SC00453J, 10.1016/j.nanoen.2020.105526;

We thank the reviewer for the suggestion, and we included the references proposed (references 19,22,54). The field is booming, and some works appeared as we were writing the review.  

  • The authors at line referred to figure 2a: perhaps they meant figure 1a;

We corrected the reference to the figure.

  • The authors used multiple references in the text: they should try to evidence the contribution of the single reference to the discussion (for example at line 164);

We fixed this issue.

  • The authors should express the measuring units more homogeneously, by using always either the apexes mode or the “/” mode (as at line 175);

We fixed this issue.

  • Perhaps in equation 6 a conversion factor is missing for having a uniform measuring unit;

We fixed this issue.

  • The sections 3.3 and 3.4 have the same title: is there an error?

Yes, it was an error, so we fixed the issue.

  • Due to the high number of symbols and acronyms, perhaps a section with their lists could be useful;

As requested by the reviewer we added the following section at the end of the manuscript: “5.Acronyms and Symbols”.

Round 2

Reviewer 3 Report

The authors well improved the paper. In my opinion, it can be published on the journal